# *UBC* Gene Family Analysis in *Salvia castanea* and Roles of *ScUBC2/5* Genes under Abiotic Stress

**DOI:** 10.3390/plants13101353

**Published:** 2024-05-14

**Authors:** Longyi Zhu, Yuee Sun, Najeeb Ullah, Guilian Zhang, Hui Liu, Ling Xu

**Affiliations:** 1Zhejiang Province Key Laboratory of Plant Secondary Metabolism and Regulation, College of Life Sciences and Medicine, Zhejiang Sci-Tech University, Hangzhou 310018, China; zly851757892@163.com (L.Z.); 15562035893@163.com (Y.S.); zhanggl0924@163.com (G.Z.); 2Agricultural Research Station, Office of VP for Research & Graduate Studies, Qatar University, Doha 2713, Qatar; nullah@qu.edu.qa; 3Faculty of Science, UWA Institute of Agriculture, UWA School of Agriculture and Environment, The University of Western Australia, Perth, WA 6009, Australia; hui.liu@uwa.edu.au

**Keywords:** *Salvia castanea*, *Salvia miltiorrhiza*, *UBC* gene family, abiotic stresses, secondary metabolite contents

## Abstract

*Salvia castanea* Diels, a relative of the medicinal plant *Salvia miltiorrhiza* Bunge, belongs to the genus *Salvia* and family *Lamiaceae*. Ubiquitin-conjugating enzyme E2 (*UBC*) is an important ubiquitin-binding enzyme in protein ubiquitination. This study aimed to analyze the regulatory role of *UBC* genes, particularly *ScUBC2/5*, on the growth and adaptation of *S. castanea* to extreme environments including cold or drought stress. We identified nine *UBC* genes in *S. castanea* and found that these genes were extremely stable and more highly expressed in the roots than other tissues. This suggested that *UBC* genes might play a role in promoting root adaptation to cold and dry environments. Further analysis of *UBC* gene expression in hairy roots under cold (4 °C) and UV stress also confirmed their importance under stress. The contents of tanshinone and salvianolic acid in hairy roots with the overexpression of *ScUBC2/5* were increased compared to non-transgenic wild type, and the cold and UV resistance of hairy roots was increased compared with that of wild type. Together, these findings highlighted the role of *ScUBC2/5* in enhancing secondary metabolite accumulation and regulation in response to cold and ultraviolet stress in *S. castanea*, providing a new perspective for genetic improvement in its phytochemistry.

## 1. Introduction

*Salvia castanea* Diels, a perennial fragrant herb with castaneous flowers, is mainly distributed in areas with an altitude of 2500–3750 m [1]. *Salvia* is a very large genus of Lamiaceae, with nearly 900 species [2], including well-cultivated Chinese traditional medicine *Salvia miltiorrhiza* Bunge, also known as “DanShen” [3]. *S. castanea* is an important wild germplasm resource in the Qinghai–Tibet Plateau, and an important relative species of *S. miltiorrhiza*. Both species can be used as medicinal plants for treating cardiovascular diseases [4]. The active ingredients of these plants can regulate menstruation by dispelling blood stasis, relieving pain, eliminating carbuncles and irritation, and nourishing the blood [5,6]. *S. castanea* has certain ornamental properties but this species has particularly been explored for its medicinal values both in China and overseas. In Tibet and some other parts of China, *S. castanea* is often used as a substitute for *S. miltiorrhisa* for treating cardiovascular diseases, palpitations, insomnia, jaundice fever, etc. [7]. With high antioxidant capacity, *S. castanea* has no adverse reactions for long-term medication usage [8].

Abiotic stress affects plant survival and growth, and their accumulation of secondary metabolites [9]. Extreme climates in high-altitude areas, such as long-term sunshine, low temperature, and high ultraviolet radiation intensity, will affect the evolution of plants. To adapt to these special plateau climates, plants have developed specialized traits, such as densely covered surfaces, fleshy leaves, colorful appearances, and a high content of secondary metabolites [10,11]. The adaptation to harsh climates in *S. castanea* is reported to be facilitated by its active ingredients, including both fat-soluble compounds, such as tanshinoneIIA (TIIA), tanshinoneI(T-I), and cryptotanshinone (CT), and water-soluble compounds, such as salvianolic acid B [12,13,14,15]. In addition, phenolic substances have an important role in protecting plants from ultraviolet radiation, forming a protective barrier against UV-B at high altitudes [16]. And terpenoids regulate plant growth and development [17]; for example, tanshinone IIA can protect cells from oxidative damage by regulating reactive oxygen species (ROS) generation [18]. Previous reports indicated significant variation in active ingredient concentrations in *S. castanea* and *S. miltiorrhiza* roots at the same developmental phase [15,19]. For example, *S. castanea* contains relatively more tanshinone, and *S. miltiorrhiza* produces higher levels of salvianolic acid B at flowering [20]. The chemistry, pharmacology, and pharmacodynamics of both species have been extensively explored, but the major divergent mechanism remains unknown [1,21,22,23].

The ubiquitin/26S proteasome pathway is a highly efficient and specific regulatory mechanism of plant protein degradation, and plays an important role in the regulation of the biosynthesis of secondary metabolites [24]. It mediates various signal transduction pathways and immune response to environmental stress [25,26]. There are three main enzymes in this pathway: ubiquitin-activating enzyme (E1), ubiquitin-conjugating enzyme (E2), and ubiquitin ligase (E3). Ubiquitin-conjugating enzyme E2 (UBC) plays a central role in this pathway and is involved in protein degradation along with E1 and E3 [27]. Ubiquitination, a key regulator of various physiological processes including DNA repair and cell cycle control [28], is integral to multiple aspects of plant biology such as hormone responses, light signaling, embryogenesis, organogenesis, leaf aging, and defense mechanisms [29], underscoring the significant role of the *UBC* gene family in orchestrating the ubiquitin/proteasome pathway in plants. Research on the *UBC* family has been documented in many plants such as *Glycine max* L. [30], *Solanum tuberosum* L. [31], and *Brassica napus* L. [32], unraveling a major role of *UBC* genes in responding to challenges such as ultraviolet irradiation and cold stress. To better understand the adaptability of *S. castanea* to the extreme high-altitude environment, characterized by intense ultraviolet light and low temperatures, this study aims to perform a comprehensive analysis of *ScUBC2* and *ScUBC5* genes in *S. castanea* for their structures, functions, and expressions under stress, and their possible roles in regulating those secondary metabolites as active ingredients of this important medicinal plant.

## 2. Results

### 2.1. Analysis of UBC Gene Family Members in S. castanea and Other Species

In this study, a total of 9 *UBC* genes were identified in *S. castanea* (named *ScUBC1*–*ScUBC9*) and 32 *UBC* genes in other species. Isoelectric point and molecular weight analysis were performed on 41 *UBC* sequences using ExPASy (Table 1). The results showed that the difference between the isoelectric point and molecular weight of the nine *ScUBCs* was not large, the isoelectric point range was 7.69–7.72, and the molecular weight range was 16,446.96–16,606.17 Da. *Andrographis paniculata* and *Amborella trichopoda* had the largest differences in isoelectric points and molecular weights compared with *S. castanea*, which were 6.03, 8441.88 Da and 8.32, 20,429.61 Da, respectively. This result preliminarily confirmed that *S. castanea* is most closely related to *S. miltiorrhiza*, and gene functions in *S. castanea* are similar to those in *Andrographis paniculata* and *Amborella trichopoda*.

The ORF sequence length of *ScUBC2* was 447 bp, encoding 148 amino acids, including 16 strongly basic (K,R), 15 strongly acidic (D,E), and 47 hydrophobic amino acids (A,I,L,F,W,V). There were also 38 polar amino acids (N,C,Q,S,T,Y). ExPASy online software predicted the molecular weight of the protein as 16,562.15 and the isoelectric point (PI) as 7.72. This indicated a basic nature of protein with 15 negatively (Asp + Glu) and 16 positively (Arg + Lys) charged residues. The instability index (II) was calculated as 46.06, which classified the protein as unstable. This analysis indicated an aliphatic index of 75.81 and a grand average of hydropathicity of −0.311. The SMART prediction results showed that there was no low complexity of the copy area. According to the online software TMHMM-2.0, the protein was predicted to have no transmembrane helical region, indicating that it could be expressed in the prokaryotic expression system.

Similar to *ScUBC2*, *ScUBC5* had an ORF sequence length of 447 bp, encoding 148 amino acids, including 16 strongly basic (K,R), 15 strongly acidic (D,E), and 47 hydrophobic amino acids (A,I,L,F,W,V). Further, it contained 39 polar amino acids (N,C,Q,S,T,Y). ExPASy online software predicted the protein molecular weight as 16,566.10 and isoelectric point (PI) as 7.72, indicating a basic nature of the protein, with 15 negatively (Asp + Glu) and 16 positively (Arg + Lys) charged residues. With an II value of 49.75, the *ScUBC5* protein was classified as unstable. The aliphatic index was 75.81 and the grand average of hydropathicity was −0.311. SMART prediction results showed that there was no low complexity of the copy area. The online software TMHMM-2.0 predicted that the protein had no transmembrane helical region, indicating that it could be expressed in the prokaryotic expression system.

### 2.2. Phylogenetic Tree Analysis of ScUBCs

The sequence alignment diagram of *ScUBCs* is shown in Figure 1. Genes within the same branch in the evolutionary relationship shared similar functions. However, some structural variations occurred during evolution. In *ScUBCs*, most protein sequences were consistent and stable, with only a few mutations. For example, in the seventh column, only *ScUBC7* and *ScUBC8* mutated into Q, and the others remained L. These mutations may be the reason for the structural and functional specificity of *ScUBCs*.

Based on the *UBC* domain sequences of 15 species, MEGA7.0 software was used to determine the optimal amino acid replacement model as JTT + G + I. A phylogenetic tree of S. castanea and 14 other species was drawn, and the species were divided into Groups A to G according to evolutionary branches. There were seven groups in total (Figure 2). *ScUBCs* and *SmUBCs* mainly clustered into the same branch. *ScUBC1* and *SiUBC1* clustered in Group A. *ScUBC2* clustered with *SmUBC1*, *SmUBC3*, and *SmUBC6* in Group B. *ScUBC5* and *ScUBC9* clustered with *SmUBC2* and *SmUBC6* in Group C; *ScUBC3*, *ScUBC4*, and *ScUBC6* clustered with *SmUBC4* in Group D. *ScUBC7* and *ScUBC8* clustered with *SmUBC5* and *SmoUBC2* in Group G. Meanwhile, no *ScUBCs* were found in Group E or Group F. The results indicated that among the investigated species, *S. castanea* was closely related to *S. miltiorrhiza*, but far related to *Arabidopsis thaliana* (L.) Heynh., *Amborella trichopoda*, and *Musa acuminata*.

### 2.3. UBC Gene Structure Characteristics of S. castanea

The 41 sequences were visualized using TBtools 11 software (Figure 3). According to phylogenetic clades, Ga-Gg was divided into seven groups. The results showed that the sequences of *ScUBCs* had Motif 1, Motif 2, and Motif 3 with most UBCs, indicating that the sequences of this gene family were conserved and functionally similar. In addition, the *AtrUBC* sequence was the longest and had Motif 7, Motif 8, and Motif 9. There were also Motif 4, Motif 5, Motif 6, and Motif 10 in *ZmUBC4*. However, Motif 1, Motif 2, and Motif 3 were not present in *ApUBC*, which suggested some differences in the function of these genes.

### 2.4. Secondary Structure Prediction and 3D Structure Prediction of ScUBC2 and ScUBC5

The prediction results of the secondary structure of *ScUBC2* using the online software SOPMA are shown in Figure 4. Among the proteins encoded by the *ScUBC2* gene, the Alpha helix accounted for 37.8% and the extended strand accounted for 17.56%. The Beta turn accounted for 6.1% and the random coil for 38.5%, indicating that the random coil structure was the skeleton of protein UBC2. Among the amino acids encoded by the *ScUBC2* gene, proline was the most abundant. The prediction results of the secondary structure of *ScUBC5* are shown in Figure 5. Among the proteins encoded by the *ScUBC5* gene, the Alpha helix accounted for 37.84%, the extended strand accounted for 17.57%, and the Beta turn accounted for 4.73%. The random coil accounted for 39.86%, indicating that the random coil structure was the skeleton of protein UBC5. The amino acids contained in *ScUBC2* (Figure 6a) and *ScUBC5* (Figure 6b) are shown in Figure 6, both of which had the highest proportion of proline.

As can be seen from Figure 7, the GMQE value of the 3D predicted structure chart reaches 0.97, close to 1. The Ramachandran figure shows that more than 99.32% of amino acid residues are in the allowed region, indicating that the protein model is of good quality (Figure 7b). In Figure 8, the GMQE value of the 3D predicted structure chart reaches 0.96, close to 1. The Ramachandran figure shows that more than 96.62% of amino acid residues are in the allowed region, indicating that the protein model is of good quality (Figure 8b).

### 2.5. ScUBC Gene Expression Pattern

In this study, based on the expression data of *ScUBCs*, a heat map was created, and nine *ScUBC* genes were divided into three categories: GI, GII, and GIII. *ScUBC8* was classified as GI; *ScUBC3, ScUBC4, ScUBC6*, and *ScUBC7* as GII; and *ScUBC1, ScUBC2, ScUBC5*, and *ScUBC9* as GIII (Figure 9). The expression patterns of the nine *ScUBC* genes were GIII > GIII > GI in the leaf, pericarp, phloem, and xylem of *S. castanea*. Meanwhile, the expression levels of *ScUBC2* and *ScUBC5* were higher than those of the other genes.

In GIII, *ScUBC5* expression was the strongest, and the expression level in roots was higher than that in leaves. The expression level of *ScUBC2* was the highest in AR2, which was increased by 75% compared to PR2. In comparison to PR3, the expression of AR3 was increased by 51%. In GII, the expression of *ScUBC3* and *ScUBC6* in AL was increased by 90% and 74%, respectively, compared to PL. In comparison to PR1, the expression of ScUBC3 and ScUBC6 in PR1 was increased by 91% and 45%, respectively. In addition, we found that *ScUBC8* was down-regulated in leaves, and expressed in different root tissues at different ages, suggesting that the function of this gene had a low effect on perennial leaf tissue. In conclusion, the expression of *ScUBCs* was up-regulated during different development stages in *S. castanea*, indicating that this family played an important regulatory role in *S. castanea* growth. Meanwhile, the up-regulation of the *ScUBC5* gene was the most obvious, followed by *ScUBC2*, which was presumed to play an important role in the adaptation of *S. castanea* to the plateau environment. Therefore, the *UBC2* and *UBC5* genes were selected for subsequent experiments to investigate their expressions in hairy roots.

### 2.6. Screening of Hairy Roots with Overexpression of ScUBC2 and ScUBC5 in S. castanea

After 20 days of culture (Figure 10), the hairy roots of the sample were interleaved into clumps, and new white roots grew out of the branches. The medium was clarified, and the medium in conical bottles was red. After PCR amplification, agar gel electrophoresis showed that the band size was correct, which confirmed that the gene had been successfully introduced into the hairy root (Figure 11). The gel electrophoresis results showed that the hairy roots were transgenic (named *ScUBC2*-1, *ScUBC2*-2, *ScUBC2*-3, *ScUBC5*-1, *ScUBC5*-2, and *ScUBC5*-3). A single line of transgenic hairy roots was selected, and after 20 days of culture, RNA was extracted and quantitative PCR was performed. The lines with a high gene expression were selected for follow-up experiments. Figure 12 shows that *ScUBC2*-1, *ScUBC*2-2, *ScUBC*5-2, and *ScUBC5*-3 have higher gene expression levels in overexpressed lines, and there is little difference in gene expression levels between wild type and empty vector, indicating that empty plasmid has little influence on the genes of the hairy root synthesis pathway.

### 2.7. Functional Verification of S. castanea UBC2/5 under Abiotic Stress

According to quantitative PCR results, the gene expressions of those increased hairy roots that were treated at 4 °C for 3 h showed up-regulation compared to those of wild-type ATCC15834 hairy roots (Figure 13a). The expression levels of those increased hairy roots after 0.5 h ultraviolet irradiation were all up-regulated compared to those of wild-type ATCC15834 hairy roots (Figure 13b).

### 2.8. Determination of Tanshinones and Phenolic Acids in Overexpressed Hairy Roots

The HPLC results showed that compared to wild-type hairy roots, the contents of dihydrotanshinone I (DT-I), cryptotanshinone (CT), tanshinone I (T-I), and tanshinone IIA (T-IIA) in hairy roots with *UBC2* gene overexpression were significantly increased, while the contents of salvianolic acid B (SAB) were slightly decreased (Figure 14).The contents of the five substances in hairy roots with *UBC5* overexpression were significantly higher than those in wild-type.

## 3. Discussion

### 3.1. Discussion of ScUBC Gene Family

At present, the *UBC* gene family has been widely identified in soybean *Glyma max* [30], *Solanum tuberosum* [31], *Brassica napus* [32], *Litchi chinensis* Sonn. [33], and other plants. Some *UBC* family members have been systematically studied in *Arabidopsis thaliana* [34], *Oryza sativa* [35], *Nicotiana tabacum* [36], and other model plants, but no similar studies have been conducted in *S. castanea*. In this study, nine *ScUBCs* from *S. castanea* were identified using gene family analysis. Ubiquitin-conjugating enzyme E2 has been reported to contain a highly conserved UBC domain consisting of about 140–200 amino acids [37]. By comparing the evolutionary tree with 14 other species, it can be seen that *S. castanea* and *S. miltiorrhiza* are the most closely related. In domain analysis, Motif 1, Motif 2, and Motif 3 were found in both *S. castanea* and other species, indicating that the *ScUBCs* had highly conserved domains during evolution. At the same time, most of the *ScUBCs* were up-regulated in roots at different ages, and *ScUBC5* was the most up-regulated gene, followed by *ScUBC2*, suggesting that they played an important role in the adaptation to the plateau environment. The ORF sequence length of both *ScUBC2* and *ScUBC5* is 447 bp, encoding 148 amino acids, and the number of strong basic amino acids, strong acidic amino acids, and hydrophobic amino acids is the same. The only difference is that *ScUBC5* has one more polar amino acid than *ScUBC2*. Both proteins are unstable and have no low complexity, no transmembrane helical region, and can be expressed in the prokaryotic expression system.

### 3.2. Discussion of the Functional Analysis of ScUBC2 and ScUBC5 under Abiotic Stress

The effects of the overexpression of *ScUBC2* and *ScUBC5* on the accumulation of secondary metabolites, as well as the adaptation to cold and UV stress, were mainly analyzed based on these two genes. First, the hairy roots that successfully introduced *ScUBC2* and *ScUBC5* genes were screened. Then, the expression levels of those hairy roots with the gene overexpression that were treated with 4 °C cold stress or ultraviolet irradiation showed a significant increase of 1.5 to 2.5 times compared with those of wild type, indicating that the stress resistance of the hairy roots with an overexpression of *ScUBC2* and/or *ScUBC5* genes was enhanced. The UBC genes play an integral role in the biological processes of plants, such as plant growth and response to stress [38]. For example, the *AtUBC2* gene plays a key role in tolerance response to UV stress and the activation of flower suppressor genes [34]. In *Zea mays*, the expression of the *ZmUBC* gene was significantly up-regulated under drought and salt stress [38]. The results of UBC gene expression in different species under stress are consistent with our findings. The expression of the *ScUBC* gene was also significantly up-regulated under UV stress and drought stress.

### 3.3. Discussion on Changes in Secondary Metabolites in S. castanea under Stress

The accumulation of secondary metabolites such as tanshinone and salvianolic acid is closely related to the anti-stress function of *S. castanea*. The accumulation of dihydrotanshinone (DT-I) in the hairy roots of *ScUBC2* overexpression was 5.32 and 4.36 times that of the wild type, and the accumulation of cryptotanshinone (CT) was 11.22 and 10.76 times that of the wild type. The accumulation of tanshinone I (T-I) and tanshinone IIA (TIIA) was 2.67 times, 3.85 times, 6.31 times, and 11.45 times that of the wild type, and the accumulation of salvianolic acid B (SAB) was lower than that of the wild type. The accumulation of cryptotanshinone and tanshinone IIA in the hairy roots of *ScUBC5* overexpression was 19.88 times, 20.91 times, 22.32 times, and 17.01 times that of the wild type, and also more than twice the accumulation of *ScUBC2*-overexpressing hairy roots. Meanwhile, the accumulation of rhizosalvianolic acid B in *ScUBC5*-overexpressed hairy roots was 6.55 times and 1.89 times that of the wild type. It can be seen that although the accumulation of secondary metabolites in the hairy roots with the overexpression of the two genes is higher than that of the wild type, that of hairy roots with *ScUBC5* overexpression is extremely high, which is consistent with the results of the expression heat map, confirming the accuracy and reliability of the experimental results in this study.

## 4. Materials and Methods

### 4.1. Identification of UBC Gene Family in S. castanea

The gene sequence of *S. castanea* was derived from transcriptome sequencing. *UBC* gene sequences from 13 species, including *Oryza sativa*, *Zea mays*, and *Arabidopsis thaliana*, were downloaded from NCBI (https://www.ncbi.nlm.nih.gov). The gene sequence data of *S. miltiorrhiza* were obtained from the National Bioinformatics Center (https://www.cncb.ac.cn). The target sequences of *S. castanea* and *S. miltiorrhiza* were initially screened using a hidden Markov model (HMM, http://pfam.xfam.org) combined with *UBC* domain (PF00179) [39]. The CD-Search (https://www.ncbi.nlm.nih.gov/cdd, accessed on 13 October 2023), Smart (http://smart.embl.de/, accessed on 13 October 2023) [40], and Pfam databases (http://pfam.xfam.org/search, accessed on 13 October 2023) were used to verify the conserved domain structure. Subsequently, to eliminate duplicate transcripts and obtain the corresponding gene sequences of proteins, we identified the members of the *S. castanea* UBC protein sequence family. Using ExPASy (http://web.expasy.org/protparam, accessed on 13 October 2023) [41] analysis of proteins, the isoelectric point (pI) and molecular weight (Mw) were obtained.

### 4.2. ScUBC2 and ScUBC5 Secondary Structure Prediction and 3D Protein Structure Prediction

The amino acid sequences of *ScUBC2* and *ScUBC5* genes were analyzed using DNAStar (https://www.dnastar.com, accessed on 16 October 2023) and DNAMAN (https://www.lynnon.com/dnaman.html, accessed on 16 October 2023). The online software TMHMM-2.0 (https://services.healthtech.dtu.dk/services/TMHMM-2.0/, accessed on 17 October 2023) was used to predict whether the protein had a transmembrane spiral region. We used the online software SOPMA (https://npsaprabi.ibcp.fr/cgibin/npsa_automat.plpage=npsa_sopma.html, accessed on 18 October 2023) for *ScUBC2/5* secondary structure prediction. According to the homologous modeling method, the 3D protein structure model prediction diagram was constructed on swissmodel (https://swissmodel.expasy.org).

### 4.3. Family Phylogenetic Tree Construction and Analysis of Gene Conserved Domain

The Clustal W analysis method was used to complete the matching sequence. The optimal amino acid replacement model was determined using MEGA7.0 Models. The maximum likelihood (ML) method was used to construct the phylogenetic tree, and the bootstrap was set to 1000 and other parameters were defaulted. The Motif structure was identified using MEME (https://meme-suite.org) [42]. The conserved motifs of 14 species such as *Arabidopsis thaliana*, *Oryza sativa*, *Zea mays*, and *S. miltiorrhiza* were obtained by MEME, and 10 motifs were set. The 41 sequences were visualized using TBtools 11 software.

The UBC gene sequences of other species, except *S. castanea* and *S. miltiorrhiza*, were downloaded from NCBI (https://www.ncbi.nlm.nih.gov). MEGA7.0 software was applied to compare the full-length sequence of the UBC gene family protein of *S. castanea*. The phylogenetic tree of the gene family was constructed using the neighbor-joining method, and the validation parameter was set to 1000 times. Clustal X-2.1-win-msi software and MEME 5.3.0 software were used to analyze the amino acid sequences of *ScUBCs* family proteins by means of sequence comparison and conserved domain analysis.

### 4.4. Mapping of ScUBC Gene Expression Map

Based on *S. castanea* transcriptome data, screening *ScUBC* FPKM values through the cloud platform of Lc-Bio Technologies Co., Ltd. (Hangzhou, China) (https://www.omicstudio.cn), heat maps were drawn.

### 4.5. Screening of Overexpressed Hairy Roots in S. castanea

The plant material *S. castanea* was collected from Yulong Snow Mountain, Lijiang City, Yunnan Province (Appendix A, https://www.tianditu.gov.cn/, Drawing approcal number: NO. GS(2019)1671, Produced by Ministry of Natural Resources). The roots were used for this experiment, and the other tissues were frozen at −80 °C.

This experiment used the DNA extraction kit Plant Genomic DNA Kit from Hangzhou Kaitai Biotechnology Co., Ltd. (Hangzhou, China). The FastPure Universal Plant Total RNA Isolation Kit (RC411) and HiScript III 1st Strand cDNA Synthesis Kit (+gDNA wiper) were all from Vazyme Biotechnology Co., Ltd. (Nanjing, China) PrimeSTAR^®^ Max DNA Polymerase High fidelity enzyme, DL2000 DNA Marker, DL5000 DNA Marker, and 6 × Loading Buffer were from Takara Biotechnology (Kusatsu, Japan). The nucleic acid dye came from Shanghai Bioscience Co., Ltd. (Shanghai, China).

The agrobacterium-mediated transient transformation has the advantages of low cost, easy operation, and high success rate, and it is used in many research fields, including gene expression detection, gene silencing, subcellular localization, protein interaction analysis, and inhibitor function identification [43]. Several single lines of hairy roots containing a wild type of agrobacterium hair-root ATCC15834, an empty vector of pCAMBIA1300-eGFP, an overexpression vector of pCAMBIA1300-*ScUBC2*, and an overexpression vector of pCAMBIA1300-*ScUBC5* were taken. The extracted DNA was used to verify the presence of the target gene in the hairy roots for subsequent tests. RNA was extracted from the selected positive strains and reverse-transcribed into cDNA, and the obtained cDNA was used for fluorescence quantitative PCR. After obtaining the results, the 2^−∆∆Ct^ method was used to calculate the two overexpressed strains selected for each strain [44]. The primers used are shown in Table 2.

### 4.6. Functional Verification of S. castanea UBC 2/5 under Abiotic Stress

The experiment materials, instruments, and methods are the same as mentioned in Section 2.4. Overexpressed hairy roots and wild-type hairy roots were cultured in conical bottles containing 6,7 V medium [45] for 20 days and were evenly divided into 5 groups; each group had 15 bottles. One group was control without any treatment, one group was treated under ultraviolet light for 0.5 h, one group was treated under ultraviolet light for 1 h, one group was placed at 4 °C for 3 h, and one group as placed at 4 °C for 6 h. There were three biological replicates for each treatment. The treated hairy root RNA was extracted and converted into cDNA for fluorescence quantitative PCR to determine its expression. The primers used are shown in Table 2.

### 4.7. Content Determination of Tanshinones and Phenolic Acids

We took 1.0 g samples from conical bottles, dried them in the oven to constant weight, and ground them into powder in a sample grinding tube. After repeated weighing, 0.02 g of each sample was taken and dissolved in a 2 mL centrifuge tube containing 1 mL 70% methanol. After ultrasound and centrifugation, the supernatant was collected with a disposable syringe and filtered through a 0.22 µm filter membrane to obtain the sample for testing. The technique was repeated three times for each treatment. The content of active components in plant tissues was determined using a Waters e2695 high-performance liquid chromatograph (Waters, Milford, MA, USA). The chromatographic conditions were a flow rate of 1 mL/min, a column temperature of 30 °C, and a sample volume of 10 µL. The tanshinones and phenolic acids were detected at 270 nm and 288 nm, respectively. The mobile phase was 0.02% phosphoric acid aqueous solution and acetonitrile, respectively, for gradient elution [46]. The content of tanshinones and phenolic acids was calculated using the measured peak area according to the standard curves (Table 3).

## 5. Conclusions

In this study, nine *UBC* genes were identified in the *Salvia castanea Diels* genome. The specificity and identity of these *UBC* genes were verified by comparing them with the *UBC* genes of other species. In this study, hairy roots of overexpressed *ScUBC2* and *ScUBC5* genes were screened for functional verification, functional analysis, and composition determination under abiotic stress. Under 4 °C cold stress and UV stress, the expression of the genes in overexpressed hairy roots increased, which proved that the introduction of the *UBC* gene could enhance adaptation to cold and UV stress in *S. castanea*. The composition of wild-type and overexpressed hairy roots was determined by means of HPLC. An increased accumulation of secondary metabolites such as tanshinone and salvianolic acid in the overexpressed hairy roots also confirmed that the introduction of the UBC gene could enhance the adaptation of *S. castanea* to cold and UV. Overall, this study provided a foundation for comprehensive understanding of the distribution, evolutionary relationship, and probable function of *ScUBC* genes and provided a molecular approach for improving the adaptability of *S. castanea* in the wild plateau environment, as well as a theoretical basis for breeding new varieties of *S. castanea* with higher stress resistance.

## Figures and Tables

**Figure 1 plants-13-01353-f001:**
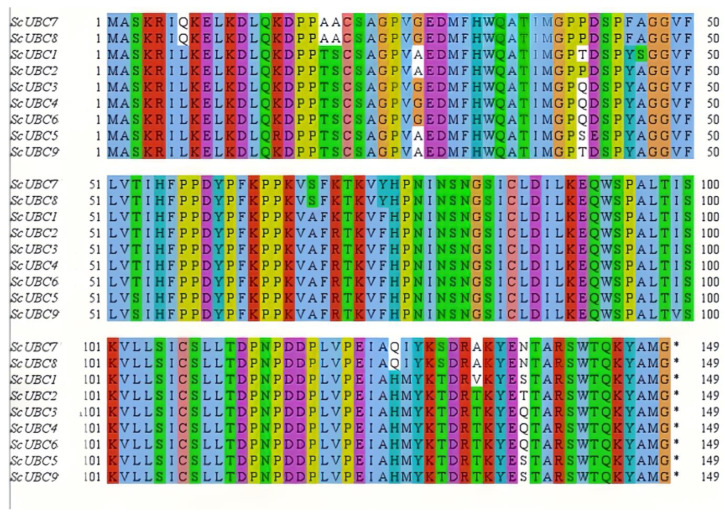
Multiple alignment of *UBC* core sequences in *Salvia castanea (ScUBCs)*.

**Figure 2 plants-13-01353-f002:**
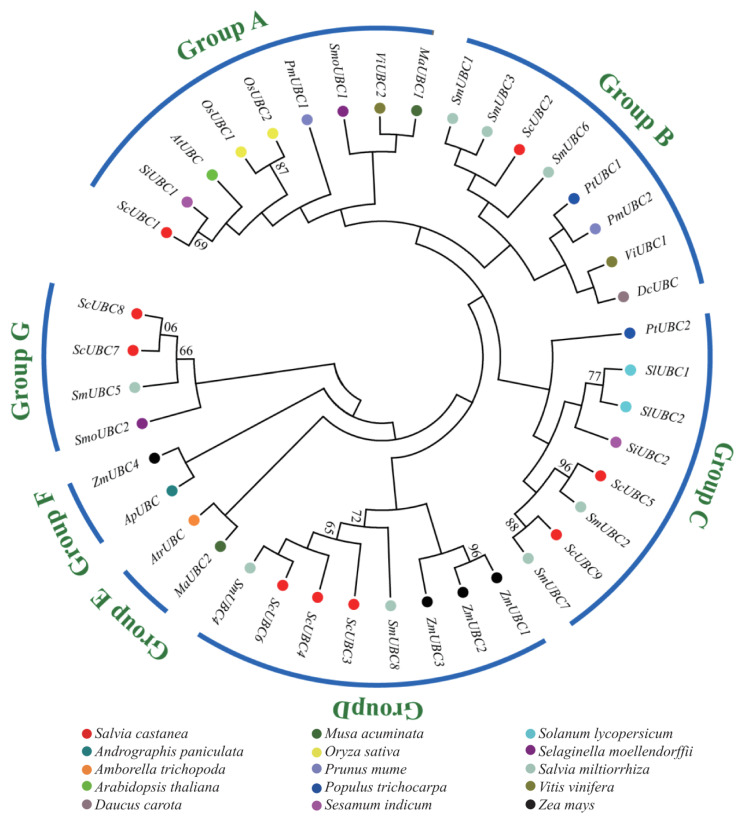
Multiple alignment of *UBC* core sequences in *Salvia castanea* and other species. Note: According to evolutionary branches, the species were divided into Groups A to G.

**Figure 3 plants-13-01353-f003:**
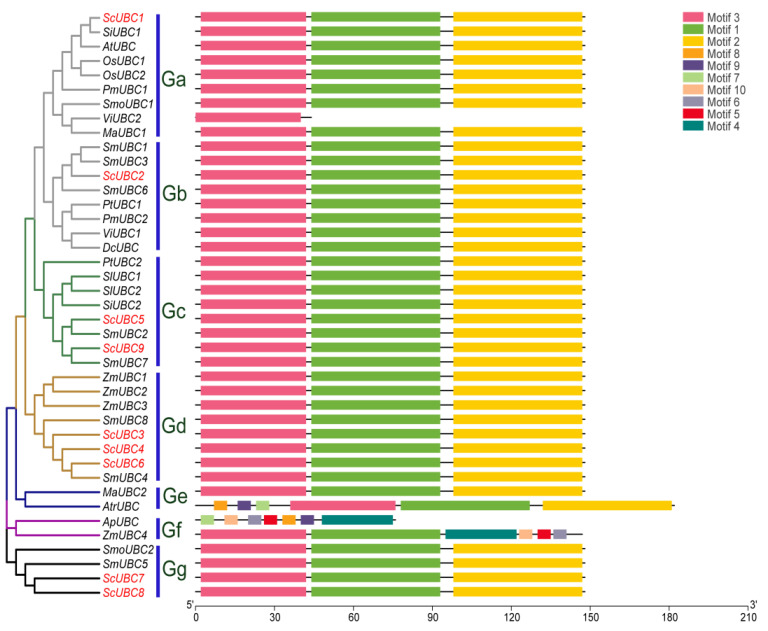
Motif structure of 41 UBC protein sequences from the 15 investigated species. Note: The red font is the UBC protein sequences of *Salvia castanea*.

**Figure 4 plants-13-01353-f004:**
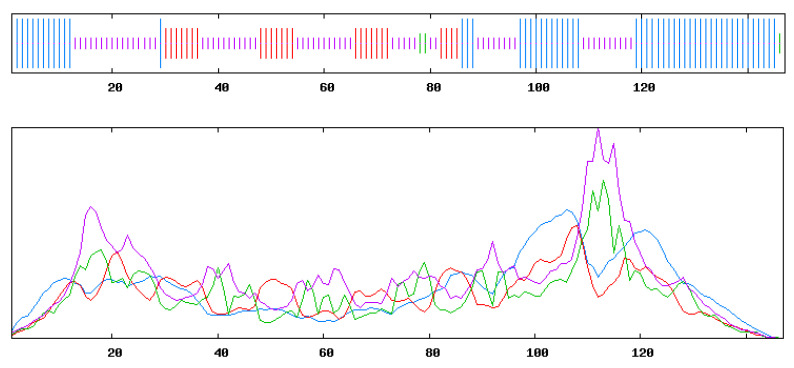
Secondary structure prediction map of protein encoded by transcription factor *ScUBC2.* Note: Blue indicates the Alpha helix; green indicates the Beta turn; purple indicates the random coil; and red indicates the extended strand. *UBC2* gene of *Salvia castanea (ScUBC2)*.

**Figure 5 plants-13-01353-f005:**
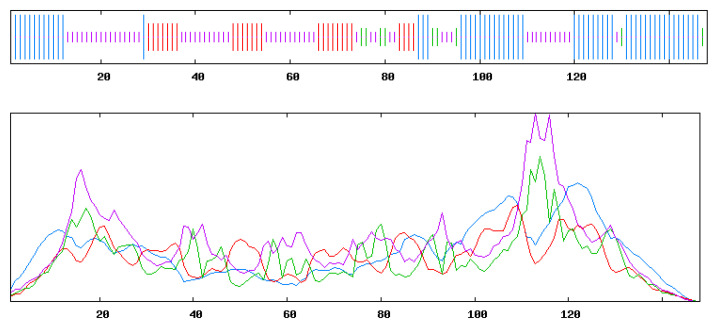
Secondary structure prediction map of protein encoded by transcription factor *ScUBC5.* Note: Blue indicates the Alpha helix; green indicates the Beta turn; purple indicates the random coil; and red indicates the extended strand. *UBC5* gene of *S. castanea (ScUBC5)*.

**Figure 6 plants-13-01353-f006:**
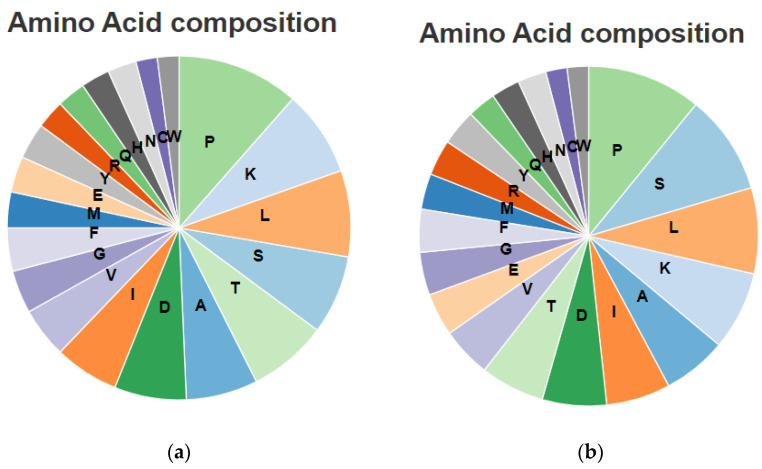
Number of each amino acid in the predicted secondary structure of the protein encoded by transcription factor genes *ScUBC2* (**a**) and *ScUBC5* (**b**) in *Salvia castanea*.

**Figure 7 plants-13-01353-f007:**
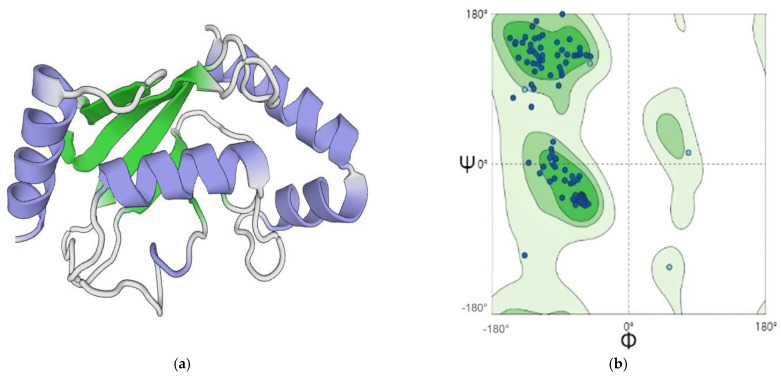
Homology modeling and model quality evaluation of *ScUBC2* in *S. castanea.* (**a**) Predicted three-dimensional structure of *ScUBC2*, with the blue ring representing the Alpha helix structure, and the green arrow representing the Beta turn structure. (**b**) Green indicates the optimal region; light green indicates the subpermissive region; and the lightest color indicates the impermissive region.

**Figure 8 plants-13-01353-f008:**
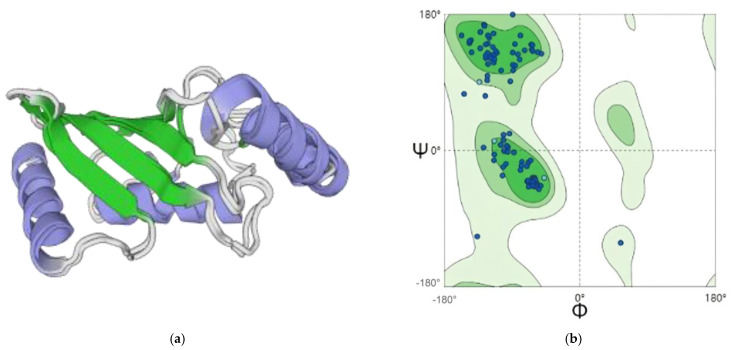
Homology modeling and model quality evaluation of *ScUBC5* in *Salvia castanea.* (**a**) Predicted three-dimensional structure of *ScUBC5*, with the blue ring representing the Alpha helix structure, and the green arrow representing the Beta turn structure. (**b**) Green indicates the optimal region; light green indicates the subpermissive region; and the lightest color indicates the impermissive region.

**Figure 9 plants-13-01353-f009:**
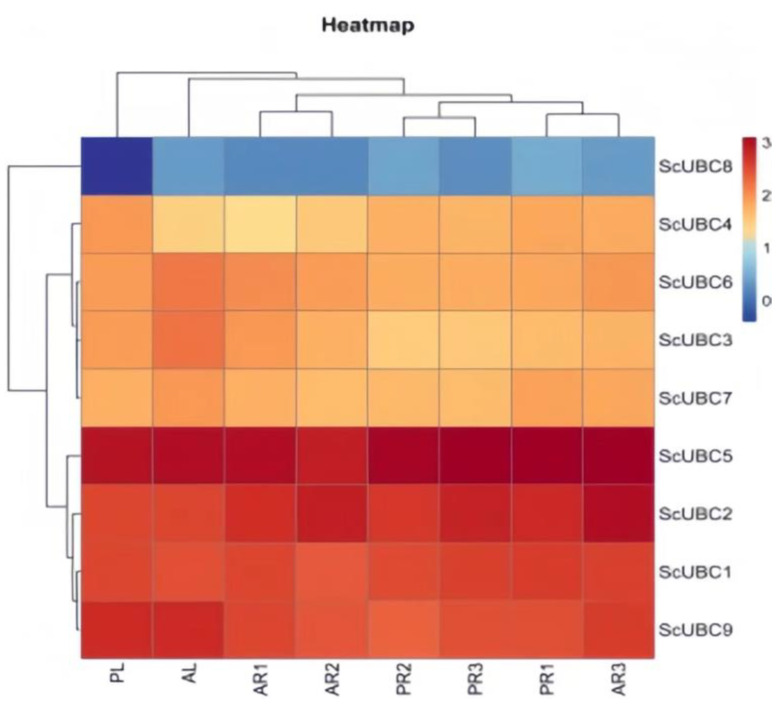
Gene expression heat map of *ScUBCs* in *Salvia castanea.* Note: AL: Annual leaf; AR1: Annual pericarp; AR2: Annual phloem; AR3: Annual xylem; PL: Perennial leaf; PR1: Perennial pericarp; PR2: Perennial phloem; PR3: Perennial xylem.

**Figure 10 plants-13-01353-f010:**
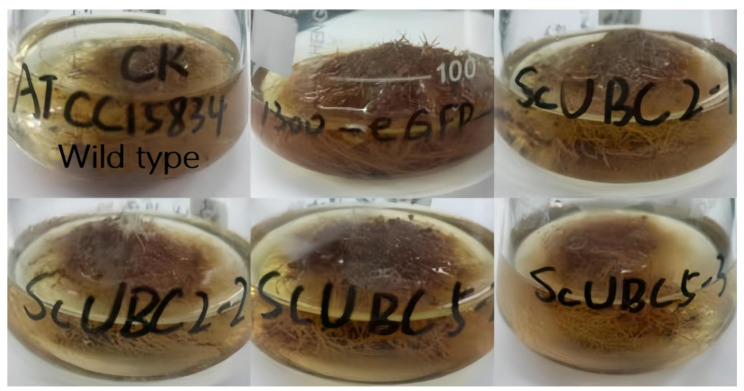
Hairy roots were cultured for 20 days.

**Figure 11 plants-13-01353-f011:**
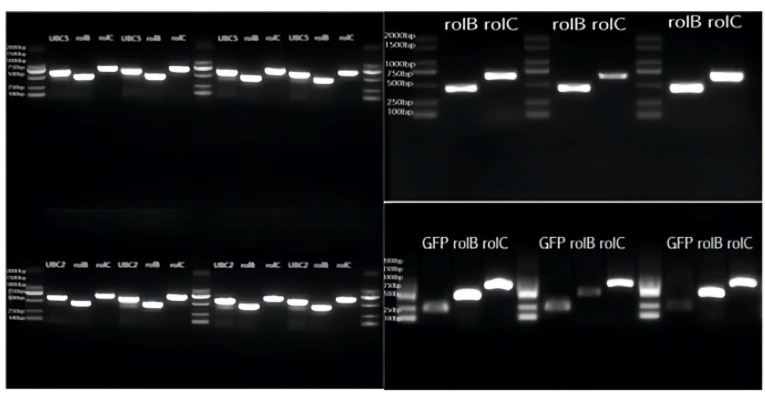
Identification of transgenic hairy roots of *Salvia miltiorrhiza* using PCR.

**Figure 12 plants-13-01353-f012:**
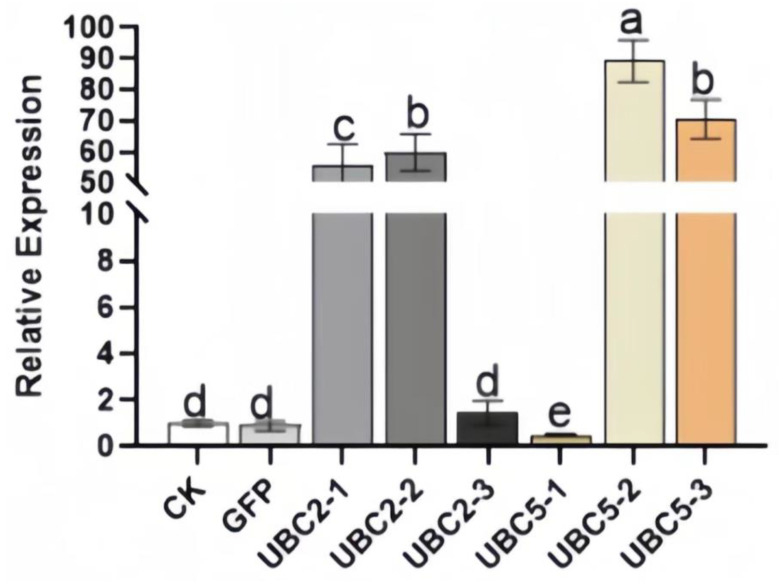
Gene expression of *ScUBC2* and *ScUBC5* in *Salvia castanea* overexpression strains. Note: All data are the average of three replicates (mean ± SD). Different lowercase letters indicate significant differences at the *p* < 0.05 level, as determined by the Duncan’s multiple range test.

**Figure 13 plants-13-01353-f013:**
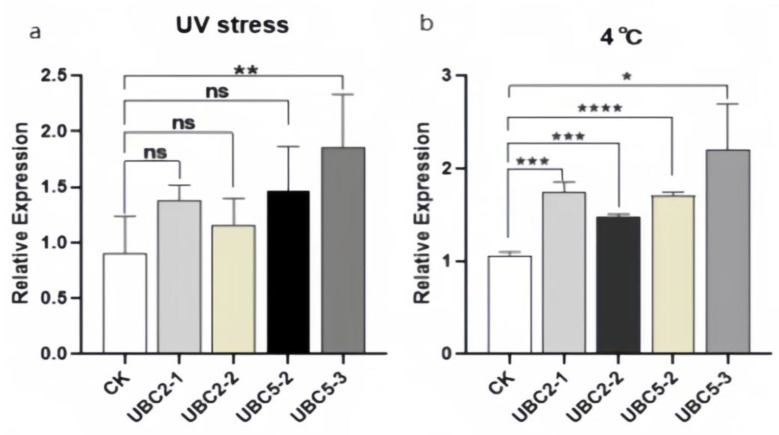
Relative expression of hairy roots after stress. Note: CK was a wild type, and the expression levels of the wild type were used as controls. (**a**) Relative expression of hairy roots after UV stress; (**b**) Relative expression of hairy roots after 4 °C cold stress. All data were the average of three replicates (mean ± SD). * *p* < 0.05; ** *p* < 0.01; *** *p* < 0.001; **** *p* < 0.0001. ns: No significant difference.

**Figure 14 plants-13-01353-f014:**
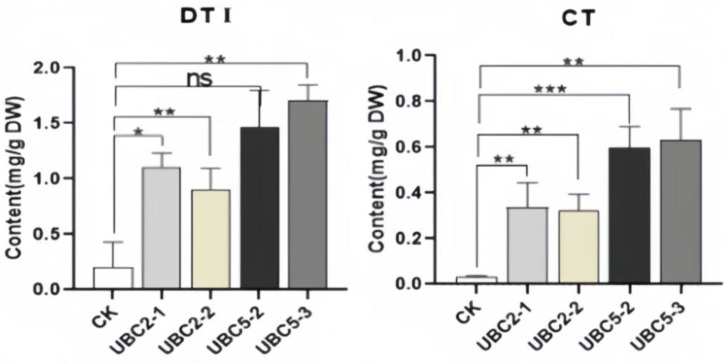
Contents of tanshinone and phenolic acids in transgenic hairy roots. Note: DT-I: dihydrotanshinone I; CT: Cryptotanshinone; T-I: Tanshinone I; T-II: Tanshinone IIA; SAB: Salvianolic acid B. A: Tanshinone I content; B: tanshinone IIA content; C: salvianolic acid B content; and D: rosmarinic acid content. CK is a wild type, and the expression levels of the wild type were used as controls. All data are the average of three replicates (mean ± SD). * *p* < 0.05; ** *p* < 0.01; *** *p* < 0.001; **** *p* < 0.0001. ns: No significant difference.

**Table 1 plants-13-01353-t001:** *UBC* gene characteristics in *Salvia castanea* and other species. The gene sequences of *S. castanea* were derived from transcriptome sequencing. The gene sequence data of *S. miltiorrhiza* were obtained from the National Bioinformatics Center (https://www.cncb.ac.cn). The *UBC* gene sequences from the other 13 species were downloaded from NCBI (https://www.ncbi.nlm.nih.gov). The abbreviated content is isoelectric point (pI) and molecular weight (Mw).

Species	Gene Name	Comment ID	pI	Mw
*Salvia castanea*	*ScUBC1*	c14380_g1	7.71	16,538.13
	*ScUBC2*	c15670_g1	7.72	16,562.15
	*ScUBC3*	c15890_g1	7.72	16,606.17
	*ScUBC4*	c15906_g1	7.72	16,606.17
	*ScUBC5*	c15861_g1	7.72	16,566.10
	*ScUBC6*	c15345_g1	7.72	16,606.17
	*ScUBC7*	c16215_g1	7.69	16,446.96
	*ScUBC8*	c16784_g1	7.69	16,446.96
	*ScUBC9*	c14548_g1	7.72	16,524.06
*Andrographis paniculata*	*ApUBC*	AFU08355.1	6.03	8441.88
*Amborella trichopoda*	*AtrUBC*	XP_006846899.3	8.32	20,429.61
*Arabidopsis thaliana*	*AtUBC*	NP_001031228.1	7.72	16,510.04
*Daucus carota*	*DcUBC*	XP_017223186.1	7.72	16,534.10
*Musa acuminata*	*MaUBC1*	XP_018679775.1	7.74	16,518.06
	*MaUBC2*	XP_018679365.1	7.72	16,530.11
*Oryza sativa*	*OsUBC1*	EAY85278.1	7.71	16,480.05
	*OsUBC2*	XP_015627363.1	7.71	16,446.03
*Prunus mume*	*PmUBC1*	XP_016651554.1	7.72	16,518.10
	*PmUBC2*	XP_008239078.1	7.71	16,521.10
*Populus trichocarpa*	*PtUBC1*	XP_002303624.1	7.72	16,522.09
	*PtUBC2*	XP_002323682.3	7.72	16,460.06
*Sesamum indicum*	*SiUBC1*	XP_011095163.1	7.71	16,510.08
	*SiUBC1*	XP_011080642.1	7.72	16,478.04
*Solanum lycopersicum*	*SlUBC1*	NP_001234247.1	7.72	16,522.09
	*SlUBC2*	NP_001294911.1	7.72	16,522.09
*Selaginella moellendorffii*	*SmoUBC1*	EFJ35297.1	7.72	16,571.16
	*SmoUBC2*	XP_024522878.1	6.82	16,797.27
*Salvia miltiorrhiza*	*SmUBC1*	EVM0014715.1	7.72	16,562.15
	*SmUBC2*	EVM0027060.1	7.72	16,566.10
	*SmUBC3*	EVM0018149.1	7.72	16,562.15
	*SmUBC4*	EVM0013655.1	7.72	16,606.17
	*SmUBC5*	EVM0026692.2	6.81	16,436.94
	*SmUBC6*	EVM0006402.1	7.72	16,562.15
	*SmUBC7*	EVM0006158.1	7.72	16,524.06
	*SmUBC8*	EVM0006069.5	7.72	16,576.14
*Vitis vinifera*	*ViUBC1*	XP_010658920.1	7.72	16,548.13
	*ViUBC2*	AAU04836.1	5.52	4857.55
*Zea mays*	*ZmUBC1*	NP_001104888.1	7.71	16,503.04
	*ZmUBC2*	AAB88617.1	7.71	16,503.04
	*ZmUBC3*	NP_001148222.1	7.72	16,507.03
	*ZmUBC4*	NP_001146962.1	7.74	16,589.06

**Table 2 plants-13-01353-t002:** Primer sequence list.

Primer Name	Sequence	Application
1-UCB2-F	TCCCGCCAGACTACCCTT	Selection of primers for *ScUBC2* overexpression lines
1-UCB2-R	TGGGTCGTCCGGGTTTGG	
2-UCB2-F	CCCGCCAGACTACCCTT	
2-UCB2-R	GGGTCGTCCGGGTTTG	
3-UCB2-F	CCATTCATTTCCCGCCAGAC	
3-UCB2-R	CAATGGGTCGTCCGGGTTT	
1-UCB5-F	AGTCCTTATGCTGGAGGTGT	Selection of primers for *ScUBC5* overexpression lines
1-UCB5-R	CGAGCAAATAGACAGCAGGAC	
2-UCB5-F	GGCAAGCCACAATCATGGG	
2-UCB5-R	GTCAATGCTGGACTCCACTG	
3-UCB5-F	GAGCAGTGGAGTCCAGCATT	
3-UCB5-R	CCCATGGCATATTTCTGGGT	
1-L-SmAOC-RT-F	CAACTCCATCCAAGGTTCAGG	Low temperature primer
1-L-SmAOC-RT-R	TCGCCGGAGTAGAGCTTGTTG	
1-PAL3-F	GTGACGTTGTGGTTGAGGAAT	Ultraviolet primer
1-PAL3-R	CTCATCAACAAGAGCAGCAAT	
2-LOX6-F	TTGAAGACTCATGCCTGCAC	
2-LOX6-R	GTCAAACCGCCACAATTTCT	
Actin-F	GGTGCCCTGAGGTCCTGTT	Internal reference primer
Actin-R	AGGAACCACCGATCCAGACA	

**Table 3 plants-13-01353-t003:** Standard curves of tanshinones and phenolic acids.

Composition	Standard Curve	Composition
DTI	y = (x − 23,027.7527)/2,660,639.56	DTI
CT	y = (x − 85,722.742)/5,526,269.802	CT
T-I	y = (x + 86,099.6853)/3,752,634.878	T-I
TIIA	y = (x + 58,317.4739)/5,128,785.876	TIIA
SAB	y = (x + 614,965.8527)/1,032,490.8342	SAB

## Data Availability

The data presented on this study are available in the article.

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
