# Peer review of "UBC Gene Family Analysis in Salvia castanea and Roles of ScUBC2/5 Genes under Abiotic Stress"

_plants, 2024, doi:10.3390/plants13101353_

Round 1
Reviewer 1 Report (Previous Reviewer 2)
Comments and Suggestions for Authors
After reviewing the corrections to the manuscript I have been satisfied with your answers and it seems to me that it is much better and could be published, just one small detail, on line 216 there is a small error in the legend of figure 8
Author Response
Dear reviewers,
Thank you for giving us the opportunity to submit a revised draft of the manuscript “Salvia castanea Diels UBC gene family analysis and effects of ScUBC2/5 gene under abiotic stress” for publication in the journal “Plants of MDPI”. We appreciate the comments and giving us a chance to improve our manuscript. All suggestions are valuable to improve our manuscript. We have read the comments carefully and revised the manuscript accordingly in track changes. The response to the reviewers’ comments and concerns are attached.
Please feel free to contact us if any further information be needed. Many thanks for your time and consideration.

Reviewer 2 Report (New Reviewer)
Comments and Suggestions for Authors
The whole introduction is a messy, logicless collection of all sorts of expressions, each time elaborated in a different way, mixing elements of ecology, phytogeography, jumping from the genus level to the species level and back again. The paper is not only logically deficient, but also lack the most elementary basic biological knowledge, and the biochemical side is also presented by jumping from individual compounds to generalisations, without any logical links. The English of the article is difficult to understand.
The authors are not aware that the introduction must end with a clearly stated aim and objectives. The authors lack experience in formulating headings for tables and figures. In many cases, statements are made without reference to literature sources. The results end with a figure. The text includes unexplained abbreviations. In the discussion section, differences in numbers are the subject of the results.
The methods are not supported by peer-reviewed sources. The authors are not familiar with the rules of citation. The description of the methods is reminiscent of laboratory notes for an undergraduate thesis. The map is inadequately prepared.
The list of references is too limited.
Comments on the Quality of English LanguagePoor English quality
Author Response
Dear reviewers,
Thank you for giving us the opportunity to submit a revised draft of the manuscript “Salvia castanea Diels UBC gene family analysis and effects of ScUBC2/5 gene under abiotic stress” for publication in the journal “Plants of MDPI”. We appreciate the comments and giving us a chance to improve our manuscript. All suggestions are valuable to improve our manuscript. We have read the comments carefully and revised the manuscript accordingly in track changes. The response to the reviewers’ comments and concerns are attached.
Please feel free to contact us if any further information be needed. Many thanks for your time and consideration.

Reviewer 3 Report (New Reviewer)
Comments and Suggestions for Authors
The manuscript titled )Salvia castanea Diels UBC gene family analysis and effects of ScUBC2/5 gene under abiotic stress( by Zhu et al. Investigated, 9 UBC genes in the Salvia castanea Diels genome. The specificity and identity of these UBC genes were verified by comparing them with the UBC genes of other species.
The paper contains interesting results and is generally well-written and structured. The experiments were successful, and the data was well understood and modeled in detail. In addition, the manuscript contains relevant paragraphs that have been discussed. The selection of the bibliography is appropriate to the content of the manuscript. However, some errors appeared throughout the manuscript, making it difficult to accept it in its current version.
- Authors should scan the manuscript for minor punctuation and English errors.
- Arrange the keywords in alphabetical order.
- All genes’ names must be in italics.
- The introduction is appropriate, but a few things need further improvements, especially the study hypothesis that should be added for the last five years.
- The title needs to be modified.
- All the gene's names must be in italics.
- The introduction is appropriate, but a few things need further improvements, especially the study hypothesis that should be added for the last five years.
- The resolution of Figures 9,12,13,14 is bad. It needs to improve.
- Conclusion: Improve this part concerning formulated objectives.
- Cross-check the references in the text and reference cite. A few references are not per the journal style in the text or reference section.
Comments on the Quality of English LanguageMinor editing of the English language is required
Author Response
Dear reviewer,
Thank you for giving us the opportunity to submit a revised draft of the manuscript “Salvia castanea Diels UBC gene family analysis and effects of ScUBC2/5 gene under abiotic stress” for publication in the journal “Plants of MDPI”. We appreciate the comments and giving us a chance to improve our manuscript. All suggestions are valuable to improve our manuscript. We have read the comments carefully and revised the manuscript accordingly in track changes. The response to the reviewers’ comments and concerns are attached.
Please feel free to contact us if any further information be needed. Many thanks for your time and consideration.

Round 2
Reviewer 2 Report (New Reviewer)
Comments and Suggestions for Authors
Salvia castanea Diels UBC gene family analysis and effects of 2 ScUBC2/5 gene under abiotic stress 3
Longyi Zhu1, Yuee Sun1, Najeeb Uallah2, Guilian Zhang1, Hui Liu3, Ling Xu1*
Ubiquitin 26 Sproteasome – error has not been improved, this shows a lack of basic knowledge in molecular biology.
Gene misspellings not corrected.
Keywords – have not been corrected according to generally accepted rules
... compared to none-transgenic wild type ... - no information is provided about the title of the main type of S. miltiorrhiza investigated in present study
Inconsistent use of the authors of the species name - it is used, not used, used again, etc.
phenolic substances ... For example, terpenoids – do terpenoids belong to phenolic substances?
Table 1. references are missing for the data included into the table
Table 1 – abbreviations are not explained in the Table 1.
Figure 2. Lots of information is missing in the Figure – species should be listed, colors should be explained, meaning of all letters in the title of the genes should be provided
Along the results there are methodical sentences which should be moved to the method chapter.
The results contain methodological sentences that should be moved to the methodology section.
In each figure Sc in the gene title should be explained
Perennial leaves , up-regulated in Annual leaves – elementary errors of botanical text..
table 2 is still BSc diploma level
In all figures and tables plant species name is missing
In discussion comparisons of the obtained data with the other results are missing.
Comments on the Quality of English LanguageSalvia castanea Diels UBC gene family analysis and effects of 2 ScUBC2/5 gene under abiotic stress 3
Longyi Zhu1, Yuee Sun1, Najeeb Uallah2, Guilian Zhang1, Hui Liu3, Ling Xu1*
Ubiquitin 26 Sproteasome – error has not been improved, this shows a lack of basic knowledge in molecular biology.
Gene misspellings not corrected.
Keywords – have not been corrected according to generally accepted rules
... compared to none-transgenic wild type ... - no information is provided about the title of the main type of S. miltiorrhiza investigated in present study
Inconsistent use of the authors of the species name - it is used, not used, used again, etc.
phenolic substances ... For example, terpenoids – do terpenoids belong to phenolic substances?
Table 1. references are missing for the data included into the table
Table 1 – abbreviations are not explained in the Table 1.
Figure 2. Lots of information is missing in the Figure – species should be listed, colors should be explained, meaning of all letters in the title of the genes should be provided
Along the results there are methodical sentences which should be moved to the method chapter.
The results contain methodological sentences that should be moved to the methodology section.
In each figure Sc in the gene title should be explained
Perennial leaves , up-regulated in Annual leaves – elementary errors of botanical text..
table 2 is still BSc diploma level
In all figures and tables plant species name is missing
In discussion comparisons of the obtained data with the other results are missing.
Author Response
Please see the attachment.

This manuscript is a resubmission of an earlier submission. The following is a list of the peer review reports and author responses from that submission.
Round 1
Reviewer 1 Report
Comments and Suggestions for Authors
This Ms needs through revision. Authors mostly did in-silico work that too with a very small gene family of merely 6 genes. Results are mostly descriptive, authors have unnecessarily elaborated the results in each section. They should follow some standard Ms for writing . Secondary and tertiary structure prediction and description and amino acid composition analyses are useless. Author should more focus on the functional data. For in-silico work, they may also include additional analysis like Ka/Ks, duplication events etc.
The Ms is very poorly written, even methodology is not properly explained.
The functional data is very limited with very poor writing. For instance, Line 247-251, what they want to say is not understandable. Simillarly most of the sections are beyond the understanding.
And most surprisingly, why there is 14 figures in this Ms? That shows authors don’t have any idea how to write the Ms.
Comments on the Quality of English LanguageEnglish very difficult to understand
Author Response
Dear Editor and Reviewers,
Thank you for giving us the opportunity to submit a revised draft of the manuscript “Salvia castanea Diels UBC gene family analysis and effects of ScUBC2/5 gene under abiotic stress” for publication in the journal “Plants of MDPI”. We appreciate the comments and giving us a chance to improve our manuscript. All suggestions are valuable to improve our manuscript. We have read the comments carefully and revised the manuscript accordingly in track changes. The response to the reviewers’ comments and concerns are given below in red.Please check the attachment for specific changes。
Please feel free to contact us if any further information be needed. Many thanks for your time and consideration.

Reviewer 2 Report
Comments and Suggestions for Authors
The analysis of the UBC gene family and the effects of the ScUBC2/5 gene under abiotic stress is the objective of this work. This species with medicinal properties has been used in places with high altitudes above sea level. In it, the authors highlight the role of ScUBC2/5 in improving the accumulation of secondary metabolites and the regulation of cold and ultraviolet stress in S. castanea, which provides a new perspective for genetic improvement in its phytochemistry.
We consider that this analysis is very original and relevant, the study of these genes can generate results that may be of importance for the development of drugs, due to their antioxidant properties and almost no side effects. The content of tanshinone and phenolic acids that can be quantified in the transgenic hairy roots is relevant.
The work is original, although in the cited literature we see a large number of works on the subject and the species, being able to demonstrate that the overexpression of this UBC gene increases the contents of tanshinone and salvianolic acid is of vital importance for obtaining transgenic plants. more tolerant to cold and UV light.
I do not consider that modifications need to be made to the methodology, most of it is from in silico approaches, I believe that the experiments were well designed and executed.
As for the conclusions, they are consistent with the results obtained, they are very precise and clear, the questions were adequately handled and a concrete answer was given.
The references are appropriate and linked to the topic, they are updated and give us an overview of the current environment of the study.
It is important to review the legends of figures 6,7 and 8
I consider that the manuscript is of importance, it is well written and also the design and results of the experiments are adequate, relevant results are shown on the topic, I believe that it can be published taking into account the details of the figure legends.
Author Response
Dear Editor and Reviewers,
Thank you for giving us the opportunity to submit a revised draft of the manuscript “Salvia castanea Diels UBC gene family analysis and effects of ScUBC2/5 gene under abiotic stress” for publication in the journal “Plants of MDPI”. We appreciate the comments and giving us a chance to improve our manuscript. All suggestions are valuable to improve our manuscript. We have read the comments carefully and revised the manuscript accordingly in track changes. The response to the reviewers’ comments and concerns are given below in red.
Please feel free to contact us if any further information be needed. Many thanks for your time and consideration.Thank you for your provite comments!
Yours sincerely,
Authours: Longyi Zhu, Ling Xu

Reviewer 3 Report
Comments and Suggestions for Authors
While this paper demonstrates strong potential, there may be room for enhancement in certain areas to align it more closely with the publication criteria of this journal. Otherwise, I have a lot of recommendations to increase the quality of your manuscript. Be careful with the writing and mistakes.
This manuscript is very interesting and is about plants adaptation to extreme climates, specifically in high-altitude areas. It mentions the active ingredients of the Salvia castanea species, which help protect it from ultraviolet radiation and other oxidative damage. The importance of phenolic compounds in protecting plants against ultraviolet radiation is also discussed. This article also includes information about the secondary structure of proteins encoded by transcription factors ScUBC2 and ScUBC5.
To enhance the article, more information could be added about the results obtained in the research, and the discussion of these results could be further elaborated.
Additionally, it would be beneficial to address the limitations of the study, providing a critical evaluation of potential constraints on the results. This would contribute to a more comprehensive understanding of the research and open the door to future investigations addressing these specific limitations.
There are several keywords repeated in the article title. In order to increase the visibility of your paper I recommend changing these keywords. If you change them by other keywords, you will increase the probability that your paper could be found by future readers when they look for your paper in some databases like Scopus for example. If you repeat the same words in the article title and in keywords, less people could find your work. So, you must think about the visibility of your research.
In the article title you must avoid the italics in the author of the species.
Line 12. You must avoid the italics in the author of the species.
Line 12. You must write the author of Salvia miltiorrhiza.
Line 30. You must delete “(S. castanea)” is redundant.
Line 31. Just before the reference [1] you must write a space. This is a very common mistake in the whole paper. Fix it. You must follow the rules of this journal.
Line 32. You must write “3750 m” instead of “3750m”. This is a very common mistake in the paper. Fix it.
Line 32. You must write a space just after a comma or a point. This is a very common mistake in your whole manuscript. Please, fix it. You must write “Tibet, Sichuan” instead of “Tibet,Sichuan”.
Line 32. You must avoid the italics in the author of the species.
Line 36. You must write a space just before the word “Both”. Just after every point you must write a space. This is a very common mistake in your manuscript.
Line 53. You must use the long hyphen between references.
Line 77. You must avoid the italics in common names of soybeans, potatoes and grapes. And you must write their scientific names with their authors because this is a botanical journal.
Line 137. The very first time that you write a scientific name of different species you must write their authors, so, you must write the authors of Arabidopsis thaliana, Oryza sativa, and Zea mays.
Line 240. Just before the word “Hairy” you must write a space. Every sentence must end with a point.
In Materials and Methods, you must write about the natural distribution of the studied plants or better create a map with their natural distribution or their origin.
All the references are wrong written. Just between the authors you must write “;”. Follow the rules of this journal.
Just download a paper of this journal and copy its style.
You must write the year of a reference just after the journal of the reference. And you must write the year in bold.
You must write at the end of all the references the doi of the references.
You must write the volume in italics.
Just between the first page and the last page of the reference you must use the long hyphen.
Otherwise, the authors adequately developed the Introduction, presenting the problems but you must write explicitly the objectives of this paper.
The methods are adequate.
The Discussion is well developed and the data presented are correctly compared with other papers.
The authors are to be congratulated for the results obtained in this article.
Comments on the Quality of English LanguageThe English is good.
Author Response
Dear Editor and Reviewers,
Thank you for giving us the opportunity to submit a revised draft of the manuscript “Salvia castanea Diels UBC gene family analysis and effects of ScUBC2/5 gene under abiotic stress” for publication in the journal “Plants of MDPI”. We appreciate the comments and giving us a chance to improve our manuscript. All suggestions are valuable to improve our manuscript. We have read the comments carefully and revised the manuscript accordingly in track changes. The response to the reviewers’ comments and concerns are given below in red.
Please feel free to contact us if any further information be needed. Many thanks for your time and consideration.
Yours sincerely,
Authours: Longyi Zhu, Ling Xu
